# Anatomical, Pathophysiological, and Clinical Aspects of Extra-Pelvic Varicose Veins of Pelvic Origin

**DOI:** 10.3390/diagnostics15030245

**Published:** 2025-01-22

**Authors:** Aleksandra Jaworucka-Kaczorowska, Marian Simka

**Affiliations:** 1Center of Phlebology and Aesthetic Medicine, 66-400 Gorzów Wielkopolski, Poland; ajaworucka@gmail.com; 2Department of Anatomy, Institute of Medical Sciences, University of Opole, 45-040 Opole, Poland

**Keywords:** pelvic vein incompetence, pelvic venous disorders, extra-pelvic varicose veins of pelvic origin, pelvic escape points

## Abstract

Venous hypertension in the pelvic veins can result in the development of varicosities in the perineum, and sometimes also in the lower extremities. These varicose veins are anatomically and functionally different from typical varicosities associated with an incompetence of the saphenous veins. Since the pelvic cavity is anatomically separated from the lower extremity and perineum by muscles and skeleton, there are only a few routes through which pelvic veins can communicate with extra-pelvic veins. These routes should primarily be examined during diagnostic workout. In this review article, clinical anatomy concerning varicose veins of pelvic origin is presented, and the anatomically-driven diagnostics for these atypical varicose veins are discussed. Focus on ultrasonographic detection of the escape points, which are located at the sites where the incompetent intra-pelvic and extra-pelvic veins are connected—such as the perineal veins, veins running alongside the round ligament of the uterus, the obturator vein, as well as the inferior and superior gluteal veins—is emphasized.

## 1. Introduction

Extra-pelvic varicose veins (VVs) of pelvic origin, both situated in the vulva (or scrotum in male patients) and in atypical locations of the lower extremity are gaining increasing clinical interest. These varicosities are one of the clinical presentations of pelvic venous disorders (PeVD) caused by pelvic vein incompetence (PVI). Based on the SVP (Symptoms, Varices, Pathophysiology) classification of PeVD, it represents S_3_V_3_P_PELV,R,NT_ [1]. These extra-pelvic VVs can be present as frequently as in 16.5% of women, with the prevalence rising to 25.6% among those with recurrent VVs. Vulvar VVs are estimated to occur in approximately 22% of pregnant and 10% of non-pregnant women, and generally in up to 34% of women presenting with enlarged and incompetent venous plexuses in the pelvis [2,3]. In addition to female patients, a number of males can also present with VVs of pelvic origin, which are typically seen in the scrotum, but in some individuals also in the lower limbs. Although the prevalence of scrotal varicose veins (VVs) is not well-documented in the literature, some studies have reported a 3% incidence of lower limb VVs of pelvic origin among male patients [4]. This rate is notably lower than that observed in females, potentially indicating pregnancy-related, hormonal, anatomical, or genetic predispositions in females. In males, however, the role of extreme exercise (e.g., excessive physical load) appears to be a critical contributing factor. Therefore, it is disputable whether PVI in women and men is really the same disease.

## 2. Pathophysiology of the Extra-Pelvic Varicose Veins

Unlike typical lower limb veins, in which pathological refluxes associated with incompetence of the saphenous and/or other superficial veins of the extremity play the primary role, it is thought that in the case of extra-pelvic VVs of pelvic origin, the congestion in the venous plexuses in the pelvis is the main etiological factor.

Pelvic VVs can result from incompetence of various pelvic veins, such as the left or right gonadal veins (ovarian or testicular veins), the left or right internal iliac veins (IIV), their tributaries, or concomitant incompetence of several veins in the pelvis. While it is typically a primary incompetence, it may also develop secondarily, due to external compression or thrombosis. For instance, compression of the left common iliac vein (CIV) by the right common iliac artery can lead to the redirection of the outflow from the lower extremity toward tributaries of the IIV, leading to pelvic venous hypertension. A similar pathomechanism, although less commonly seen, begins with the compression of the left renal vein (LRV), which is squeezed between the aorta and the superior mesenteric artery, resulting in redirecting of the venous outflow from the kidney toward the left ovarian vein, contributing to pelvic venous hypertension. Extrinsic compression may also result from such pathologies as endometriosis or tumors. Similar pathomechanisms are encountered in the settings of post-thrombotic iliac obstruction [1]. Incompetence of inferior vena cava (IVC) or iliac veins may lead to an increased venous pressure in the pelvic venous plexuses: ovarian, uterine, vesical, or prostatic. In women, these flow disturbances result in clinical symptoms, such as chronic pelvic pain and/or menstrual disorders [5]. It has been postulated that in the setting of pelvic congestion, veins connecting the intra- and extra-pelvic networks through the so-called pelvic escape points located in the pelvic floor may enlarge and their valves may become incompetent, which in turn may lead to the transmission of venous refluxes to the perineum, thighs and gluteal region. Consequently, vulvar or scrotal VVs and/or atypical lower extremity VVs can develop [6].

Pelvic vein incompetence (PVI) is influenced by several predisposing factors. One of the most important factors is pregnancy. PVI is still rare during the first pregnancy. It typically emerges around the fifth month of the second pregnancy, with an increased risk during subsequent pregnancies. Elevated levels of estrogen, progesterone and relaxin during pregnancy are thought to weaken vein walls and cause dilation [7,8,9]. Additionally, an increased blood volume and up to a 60-fold rise of the ovarian vein vascular capacity, which may persist for months after delivery, may play a role. Compression of the pelvic veins by the pregnant uterus can further contribute to this pathomechanism [10]. Additionally, anatomical variations of venous valve distribution may play a role in the development of PVI. The number of valves is higher in distally located veins. Notably, valves are found only in 10% of the IIVs. In the ovarian veins, valves are primarily located in their distal third, while 15% of the left and 6% of the right ovarian veins are devoid of valves [11].

Based on the literature, among patients presenting with extra-pelvic VVs of pelvic origin, less than 10% have been reported to have pelvic symptoms related to PVI, as the pelvic venous hypertension is transmitted from the pelvis to the distal areas, such as the veins of the vulva and lower limb [12]. For example, Gibson et al. found that only 7% of patients with extra-pelvic VVs of pelvic origin reported symptoms associated with PVI [13]. On the other hand, these patients complain of local symptoms, either localized in the external genitalia or the lower limb, comprising pain, discomfort, tenderness, itching, bleeding, or thrombosis of the varicosities. The presence of the vulvar VVs can also be associated with dyspareunia and vulvodynia [1]. For the time being, it remains unclear if VVs of pelvic origin develop in the setting of typical anatomy or whether anatomical variants are also relevant. Research on these extra-pelvic VVs of pelvic origin, despite their clinical importance, is still incomplete. This particularly concerns detailed anatomical studies, which, for obvious reasons, cannot rely on traditional methodology, including cadaver dissections, but rather have to be based on diagnostic imaging studies, primarily on ultrasonography. These caveats will be further discussed in our paper.

## 3. Anatomy of the Pelvic Veins

The CIVs are the main venous channels draining the pelvis. These veins are asymmetric, with the right vein being shorter and running more vertically, which reflects their embryological development. The left CIV vein is typically crossed by the overriding right common iliac artery, which can create a hemodynamically relevant compression (May–Thurner syndrome). The ascending lumbar veins that drain into the CIVs provide the connection with the azygos vein system. These connections provide an alternative outflow route from the lower part of the body in the case of obstruction or aplasia of the iliac veins or the IVC. The ascending lumbar veins are better developed on the left side, which reflects their embryological development [5,14].

The CIV develops as a connection of the internal and external iliac veins. The external iliac veins provide the venous outflow from the lower extremity, while the IIVs primarily drain the walls and organs of the pelvis. Quite commonly, the IIV is not a single blood vessel; there are two or more veins in this location. Anatomically, the tributaries of the IIVs comprise three categories of veins: those draining the lateral walls of the pelvis, those draining its posterior wall (particularly the sacral and coccygeal bones together with adjacent structures), and venous plexuses providing outflow from the visceral organs of the pelvis (the bladder, the uterus and vagina, the prostate, and the rectum). The first group of tributaries is of particular interest in our paper and will be further discussed. Notably, the uterine venous plexus, which primarily drains into the IIVs, is also connected through veins of the broad ligament of the uterus with the ovarian veins, which ultimately drain to the LRV (left side) or the IVC (right side). These connections can also be clinically relevant, especially in patients presenting with a narrowed LRV. The uterine venous plexus is also connected with the veins running through the inguinal canal, alongside the round ligament of the uterus. Through these veins, venous hypertension can be transmitted from the venous plexuses of the pelvis to the veins in the groin [15].

There are numerous venous tributaries of the IIV that drain the lateral walls of the pelvis. Those of particular clinical relevance comprise the superior and inferior gluteal veins, the internal pudendal veins, and the obturator veins. All these veins, in addition to the above-mentioned veins of the round ligament of the uterus, and to the vaginal and pudendal venous plexuses, can be anatomically associated with the varicosities of pelvic origin [16,17,18].

Although the pelvis is primarily drained by the IIVs, in women, venous outflow from this area is also directed to the ovarian veins. The left ovarian vein is typically a tributary of the LRV, while the right ovarian vein joins the IVC. In men, a similar pattern is exhibited in the testicular veins; still, these veins primarily drain the testes, not the pelvic organs. The proximal part of the LRV is located between the aorta and the superior mesenteric artery. Here, especially in slim individuals, this vein can be compressed, which may lead to an impaired venous outflow from the left kidney, which in turn can result in an overload of the left ovarian vein. This compression can result from a small angle between the aorta and the superior mesenteric artery, but can also be temporary, such as during pregnancy or following a significant weight loss. Other anatomical abnormalities of the LRV, including the rare LRV hypoplasia or the more common retroaortic course of the LRV or the left ovarian vein draining into the retroaortic branch of the LRV, should also be considered.

Since the ovarian veins communicate with the vascular territory of the IIVs through the venous plexuses of the broad ligament of the uterus, compression of the LRV can finally result in dilatation of the pelvic veins or even in the development of extra-pelvic varicosities [1,5].

## 4. Anatomy of Pelvic Escape Points and Related Extra-Pelvic VVs

The term “pelvic escape points” refers to specific anatomical connections between the pelvic veins and veins of the perineal region and/or veins of the lower extremities. These connections were first described and categorized by Franceschi [19]. Medical literature typically describes seven to eight such pathways. Understanding these escape points is essential for the diagnostic workout and treatment of extra-pelvic varicosities of pelvic origin [17,19,20,21,22,23].

Although pelvic escape points are primarily assessed from the perineum and groin, their anatomy from the pelvic side should also be acknowledged. Importantly, venous plexuses in the pelvis are separated from the veins located in the perineal, inguinal, and gluteal areas by the sacrum and pelvic bones, and associated muscles, fibrous membranes, and ligaments. In addition to the main physiologic pathway, which is located under the inguinal ligament, there are only a few narrow routes providing connections between the intra- and extra-pelvic veins in the settings of pathology.

These additional pathways can become dilated, particularly during pregnancy, and can facilitate the development of extra-pelvic VVs. In the females presenting with normal anatomy, there are four such openings in the floor of the pelvis: the lesser sciatic foramen, where the internal pudendal veins enter the pudendal canal, the greater sciatic foramen (particularly its infrapiriform part), the deep inguinal ring, and the obturator foramen (Figure 1). Out of these four locations, the lesser sciatic foramen, which facilitates communication alongside the internal pudendal veins, is of particular clinical interest, because a majority of VVs of pelvic origin are associated with this opening in the pelvic floor. Venous refluxes originating at the greater sciatic foramen are often associated with anatomical variants resulting from incomplete rebuilding of the fetal pattern of the lower limb veins. The route through the inguinal canal is primarily associated with venous hypertension in the pelvis. The route through the obturator foramen is probably the least understood. The obturator veins join the tributaries of the profunda femoris vein (the deep vein of the thigh). All these veins are situated inside the medial fascial compartment of the thigh. Current anatomical textbooks do not describe the important connections of these veins with the superficial veins in the groin or perineum. On the other hand, VVs associated with incompetent obturator veins are well known. Obviously, more good quality research is needed regarding this topic.

Acknowledging all the above-described anatomical details concerning pelvic veins and openings in the pelvic walls, diagnostic workout for extra-pelvic VVs should primarily focus on revealing the external escape points located in the perineum, inguinal area, or below the buttocks. This diagnosis should be performed by means of ultrasonography, typically utilizing a linear sonographic probe [2,24,25]. In patients with coexisting pelvic symptoms, possibly of venous etiology, when a pelvic vein intervention is considered, the initial diagnosis of PVI, including ultrasonography, should be further verified by detailed imaging, using either magnetic resonance venography or computed tomography [6].

### 4.1. Pudendal Escape Points

The pudendal pelvic escape points are the most common and are found in 60–70% of women presenting with varicosities of pelvic origin. Refluxes associated with these escape points originate in the internal pudendal veins, which are tributaries of the IIV. The internal pudendal veins traverse the pelvic floor through the lesser sciatic foramen and then run alongside the pudendal canal (Alcock’s canal) in proximity to the ischial spine and inferior ramus of the ischium. External connections associated with incompetent internal pudendal veins are located where these veins traverse the anterior part of the perineal membrane (fibrous sheath separating the pelvic cavity from the perineum). Here, communication between pelvic venous plexuses with the superficial veins of the urogenital area becomes possible. Through this pathway, venous reflux from the pelvis can be transmitted to such superficial veins of the perineum as the deep dorsal veins of the clitoris, the veins of the bulb of the vestibule, the perineal veins, and the inferior rectal veins (Figure 2) [6,26,27,28]. In men, the relevant veins comprise the deep dorsal veins of the penis, the veins of the bulb of the penis, the perineal veins, and the inferior rectal vein (Figure 3).

Anatomically, the pudendal pelvic escape points can be categorized into three groups: the clitoral, the intermediate labial, and the perineal [26]. Refluxes that originate in the internal pudendal veins can also be transmitted to the inferior rectal veins, although such refluxes are rarely found in patients presenting with venous congestion in the pelvis. Dilated venous external rectal plexuses rather suggest the presence of portal hypertension [29].

#### 4.1.1. Clitoral Escape Point

The clitoral escape point is located on both sides of the clitoris and consists of anastomotic venous plexuses connecting the external pudendal veins (superficial and deep) with the internal pudendal veins. In this area, veins located above the perineal membrane (drained by the internal veins) connect to the veins located below this membrane. The latter veins belong to the territory of the external pudendal veins. The deep dorsal vein of the clitoris, the tributary of the internal pudendal veins, connects to the superficial dorsal veins of the clitoris, which are tributaries of the external pudendal veins. The external pudendal veins have a short course that passes superomedially from the clitoris and the anterior part of the labium majus and drains to the great saphenous vein (GSV) distally from the saphenofemoral junction (typically, between the terminal and pre-terminal valves of this vein) (Figure 4). Through this pathway, reflux originating in the pelvis, via the internal pudendal veins, can be transmitted to the superficial veins of the perineum, and even downwards, to the lower extremity veins. Reflux coming from the clitoral escape point can result in the development of varicosities at the mons pubis, in the inguinal area, atypical VVs at the anterior aspect of the thigh, as well as incompetence of the GSV and/or the anterior accessory saphenous vein (AASV), together with varicosities draining to these interfascial veins [26,28,30]. A characteristic feature of such VVs located in the thigh is that an incompetence of these veins begins below the connection with the external pudendal veins. Consequently, the terminal valve of the GSV is competent. It should be remembered that proper detection of the venous reflux across the saphenofemoral junction (SFJ) in such cases should not rely on the Valsalva maneuver, which can be misleading. Instead, reflux should be provoked by the compression/release of the GSV or dilatated veins in the upper thigh, with the Doppler gate located in the femoral vein just proximally from its junction with the GSV [31]. Patients with a clitoral pelvic escape point may also develop vulvar VVs through tributaries of the external pudendal veins: the anterior labial veins (Figure 5). It should be noted that due to venous connections, the ipsilateral clitoral escape point may additionally cause contralateral extra-pelvic VVs of pelvic origin [5,26,32].

The ultrasonographic appearance of the atypical VVs, located at the anterior aspect of the thigh and associated with the clitoral pelvic escape point, can be seen in the following multimedia file (Appendix A).

In males, although much less frequently than in females, a similar reflux pattern can develop [26]. Venous reflux originating in the internal pudendal veins can be transmitted to the deep dorsal veins of the penis, which are tributaries of the internal pudendal veins (Figure 3a). Through the connections of the deep dorsal veins of the penis with the superficial dorsal veins of the penis (tributaries of the external pudendal veins), reflux can be transmitted to the scrotum, resulting in scrotal varicosities (Figure 6). It should be remembered that usually scrotal VVs are not associated with incompetent testicular vein. Such an incompetence can result in varicocele (dilated pampiniform plexus), but not VVs of the scrotum [33]. Similarly to females, lower limb VVs, instead of incompetent valves of the GSV and/or AASV, can be associated with the escape point associated with the deep dorsal veins of the penis, an equivalent to the female clitoral escape point.

#### 4.1.2. Intermediate Labial Escape Point

The intermediate labial escape point is an alternative pathway for the reflux that originates in pelvic veins and can be transmitted to the veins of the perineum and lower limb. This pathway is anatomically related to the veins of the bulb of the vestibule, which are tributaries of the internal pudendal veins (Figure 2). The veins of the bulb of the vestibule are located within the vestibular bulb, on either side of the vaginal opening. They originate around the vestibule of the vagina, in the superficial perineal space, extend along the anterior or lateral vaginal wall and pierce the perineal membrane to drain into the internal pudendal veins [26]. As terminal branches, they receive no tributaries but may be responsible for VVs of the labia majora and/or medial aspect of the thigh. Figure 7 and Figure 8 present ultrasonographic and clinical pictures of VVs associated with this pelvic escape point.

#### 4.1.3. Perineal Escape Point

The perineal escape point is perhaps the most frequently observed pudendal pelvic escape point. Venous reflux coming from the pelvis is transmitted via the internal pudendal veins, and subsequently through the perineal veins, tributaries of the internal pudendal veins (Figure 2 and Figure 3). Through this pathway, venous reflux can be transmitted from the pelvis to the tributaries of the perineal veins: the posterior labial veins in females, or the posterior scrotal veins in males, resulting in the development of extra-pelvic VVs. Incompetence of the perineal veins can lead to the development of vulvar or scrotal VVs. In addition, these veins connect to the superficial veins of the lower limb, resulting in atypical VVs located at the anterior, medial, or posterior aspect of the thigh. Typically, in such cases the terminal valve of the GSV is competent. Notably, the veins of both labia majora are interconnected. Therefore, venous reflux originating in the perineal veins on one side, can be transmitted to the contralateral labium majus [5,26].

In females (Figure 9), the perineal veins exit the superficial perineal fascia at the distal third of the labium majus, approximately 1 cm from the origin of the frenulum of the labium minus. They run along the posterior vaginal wall, in the space between the vagina and the rectum (Figure 9), traverse the perineal muscles and the perineal membrane, and ultimately join the internal pudendal veins within the pudendal canal [26,28,30,31,32].

In men, the perineal veins and the posterior scrotal exhibit a similar topography. The posterior scrotal veins form a venous plexus in the skin and connective tissue of the scrotum, run along the posterior aspect of the scrotum, and form the perineal veins, which join the internal pudendal veins anteriorly from the rectum (Figure 3 and Figure 10).

### 4.2. Inguinal Escape Point

The inguinal escape point is the second most common escape point, following the pudendal one. It is found in 21–36% of patients with extra-pelvic VVs of pelvic origin [26,27]. Seen from the groin, venous reflux related to this escape point originates at the superficial inguinal ring. In females, reflux is linked to dilated and incompetent veins of the round ligament of the uterus, which traverse the inguinal canal (Figure 11a) and connect to venous plexuses near the uterus, particularly within its broad ligament. While commonly related to ovarian vein incompetence, this pelvic escape point can also result from the incompetence of other pelvic veins. In males, the inguinal pelvic escape point rarely results in superficial varicosities. It is rather associated with the incompetence of the testicular vein and its distal extension, the pampiniform plexus, resulting in varicocele [34]. Patients presenting with reflux originating in the inguinal pelvic escape point can develop ipsilateral or contralateral vulvar/scrotal or perineal VVs. Additionally, through connections with the external pudendal veins and branches of the superficial perineal network, lower limb VVs can also occur (Figure 11) [26]. This pelvic escape point quite often mimics the clinical presentation of an inguinal hernia, and requires an ultrasonographic investigation for accurate diagnosis.

The ultrasonographic appearance of the atypical VVs, located at the anterior aspect of the thigh and associated with the inguinal escape point, can be seen in the following multimedia file (Appendix A).

### 4.3. Gluteal Escape Point

These pelvic escape points are associated with the incompetence of the superior or inferior gluteal veins, tributaries of the IIVs. The superior gluteal veins enter the pelvic cavity through the suprapiriform part of the greater sciatic foramen, while the inferior gluteal veins pass through the infrapiriform one. Both veins connect with numerous superficial veins in the gluteal and subgluteal regions, including gluteal perforating veins and their subcutaneous tributaries [26,28,30,31]. Incompetence in these veins can lead to extra-pelvic VVs of pelvic origin in these areas (Figure 12). The inferior gluteal escape point accounts for 3.7% of VVs of pelvic origin, whereas the superior gluteal escape point is responsible for 1.6% [27]. The inferior gluteal veins accompany the sciatic nerve as they enter the pelvic cavity. In the distal thigh, these veins connect to the veins of the sciatic nerve. This topography is a remnant of the fetal pattern of the venous system of the lower extremity, where the axial vein (which later transforms into the veins of the sciatic nerve) is the main vein of the limb and accompanies the axial nerve (which later becomes the sciatic nerve) [35]. In some individuals, this fetal pattern persists. Notably, the veins of the sciatic nerve communicate with the profunda femoris vein (the deep vein of the thigh), offering an alternative outflow from the lower limb to the inferior gluteal veins in cases of femoral vein obstruction or hypoplasia. While most of the varicosities in the buttocks or the posterior upper thigh are due to pelvic venous hypertension, some are linked to these embryological remnants.

### 4.4. Obturator Pelvic Escape Point

This escape point is the least commonly encountered pelvic escape point and is responsible for 3.2% of extra-pelvic VVs of pelvic origin [35,36]. It is associated with incompetent obturator veins, the tributaries of the IIV. The obturator veins enter the pelvis through the upper part of the obturator foramen (Figure 13). Reflux from pelvic veins is transmitted through the obturator veins to the tributaries of the profunda femoris vein (the deep vein of the thigh) located within the medial fascial compartment of the thigh (primarily, the veins of the adductor muscles). Through the circumflex femoral veins, connections with other fascial compartments of the thigh are also provided. Besides, the obturator veins connect to the internal pudendal veins and the inferior epigastric veins. VVs associated with incompetent obturator veins are typically located in the perineum and medial aspect of the upper thigh [26,28,30,31]. VVs resulting from incompetent obturator veins are typically seen in the perineum and the medial upper thigh (Figure 13). Notably, the obturator veins serve as a significant outflow pathway for the lower limb in cases of thrombotic occlusion of proximal part of the femoral vein or the external iliac vein.

## 5. Conclusions

There are several pathways through which pelvic venous reflux can be transmitted to the veins of the vulva, scrotum, perineum, and the lower extremity, leading to the development of extra-pelvic VVs of pelvic origin. Since the clinical appearance of these varicosities is similar, anatomical knowledge of the connections between intra- and extra-pelvic veins is essential in the context of proper diagnosis and treatment [13,36,37]. Detection of all active escape points allows for adequate management of these atypical VVs and improves the results of the treatment, with reduced recurrence rates of these difficult-to-treat varicosities. However, detailed anatomy and functional variability of the pelvic floor veins remain poorly understood. It seems that anatomical variations of these veins and nearby structures, rather than hemodynamics alone, may play the primary role in the development of these extra-pelvic VVs, particularly in asymptomatic females. Further research is needed to investigate this topic more thoroughly.

## Figures and Tables

**Figure 1 diagnostics-15-00245-f001:**
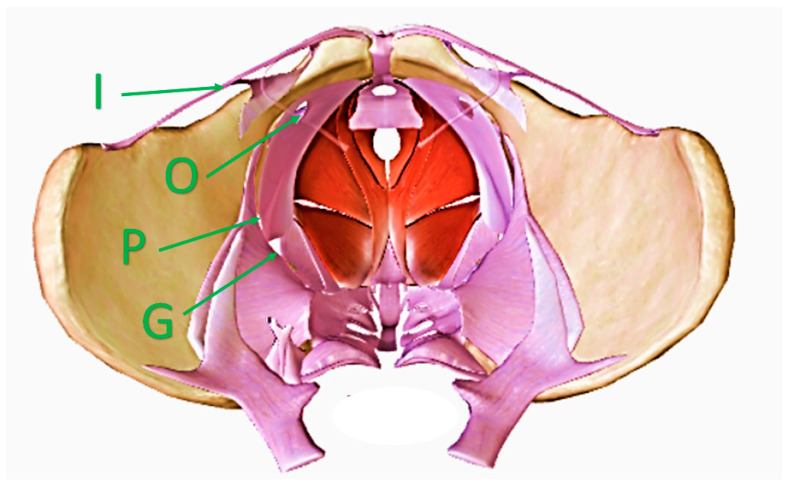
Female pelvis seen from above, with openings in the pelvic floor and veins providing communication with the perineum, gluteal area, and the lower limb. I—the inguinal canal through the veins of the round ligament of the uterus; O—the obturator foramen through the obturator veins; P—the lesser sciatic foramen through the internal pudendal veins; G—the greater sciatic foramen through the gluteal veins.

**Figure 2 diagnostics-15-00245-f002:**
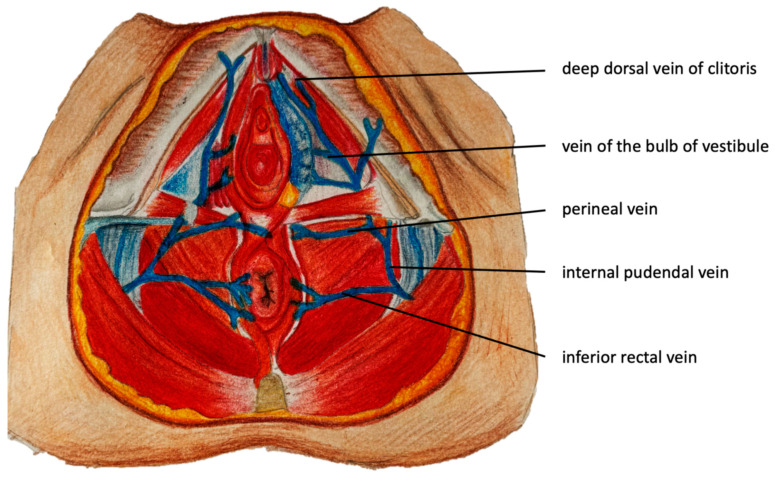
Transverse section of the female perineum with tributaries of the internal pudendal veins.

**Figure 3 diagnostics-15-00245-f003:**
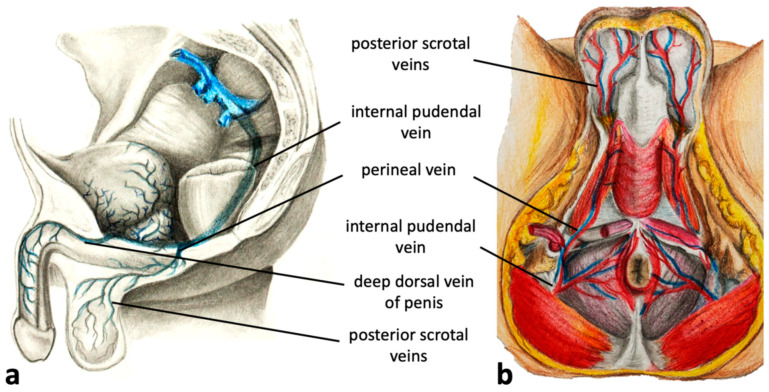
(**a**) Sagittal and (**b**) transverse sections of the male perineum with tributaries of the internal pudendal veins.

**Figure 4 diagnostics-15-00245-f004:**
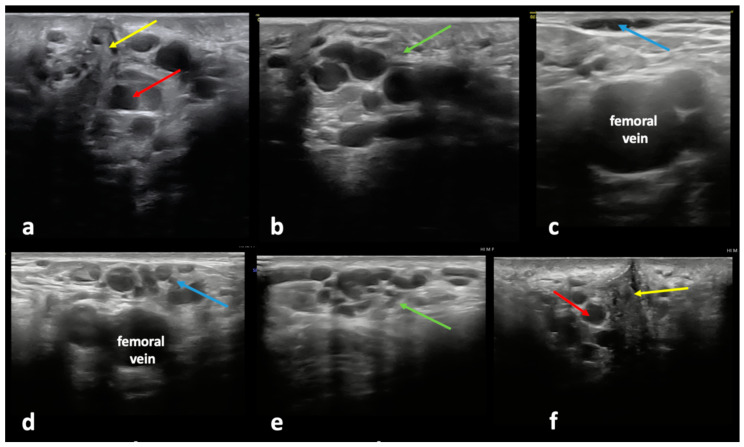
Ultrasonographic assessment of the clitoral pelvic escape point and related varicose veins (VVs): incompetent clitoral pelvic escape point (red arrow) on the (**a**) left and (**f**) right side of the clitoris (yellow arrow), resulting in (**b**,**e**) a dilated and incompetent external pudendal vein (green arrow) forming VV clusters at the mons pubis and the inguinal area, causing (**c**,**d**) atypical VVs (blue arrow) at the anterior aspect of the thigh.

**Figure 5 diagnostics-15-00245-f005:**
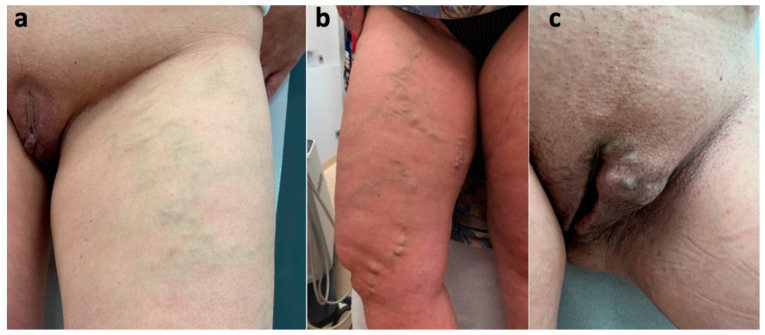
Varicose veins in female patients caused by an incompetent clitoral escape point and reflux extending to the external pudendal veins: (**a**) atypical varicose veins at the anterior aspect of the thigh; (**b**) varicose veins related to the incompetent anterior accessory saphenous vein; (**c**) vulvar varicosities associated with incompetence of the anterior labial veins, tributaries of the external pudendal veins.

**Figure 6 diagnostics-15-00245-f006:**
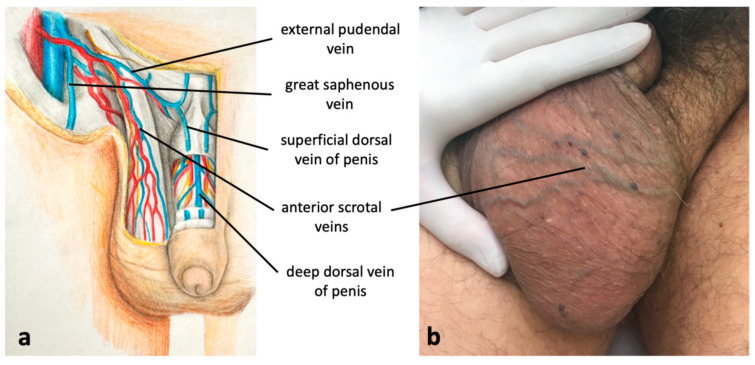
(**a**) Connections between pelvic veins (the deep dorsal vein of the penis, tributary of the internal pudendal veins) and superficial veins of the scrotum and lower extremity; (**b**) scrotal varicosities caused by incompetent internal pudendal veins, connected to incompetent external pudendal vein and anterior scrotal veins.

**Figure 7 diagnostics-15-00245-f007:**
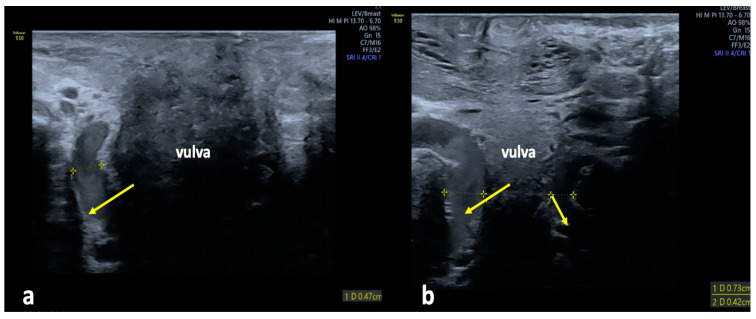
Ultrasonographic assessment of the intermediate labial escape point in females presenting with extra-pelvic VVs of pelvic origin; (**a**) longitudinal view of an enlarged vein of the vestibular bulb (arrow) extending along the anterior wall of the vagina; (**b**) transverse view of an enlarged vein of the vestibular bulb (arrows) located in the space anteriorly to the vagina.

**Figure 8 diagnostics-15-00245-f008:**
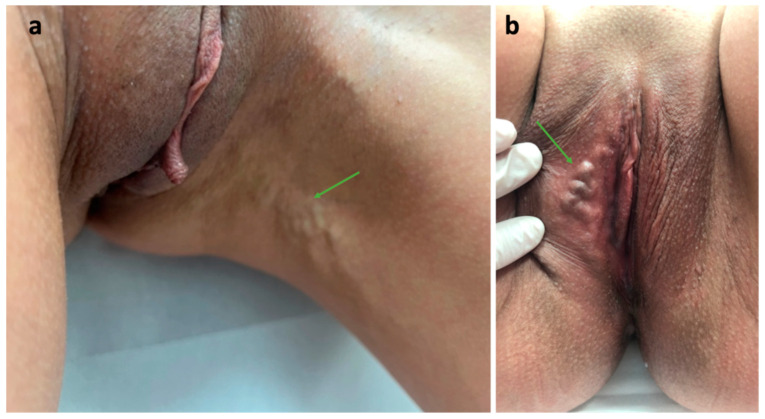
Varicose veins related to the incompetent intermediate labial escape point; (**a**) atypical varicosities (arrow) at the medial aspect of the thigh; (**b**) varicosities (arrow) in the vulvar area.

**Figure 9 diagnostics-15-00245-f009:**
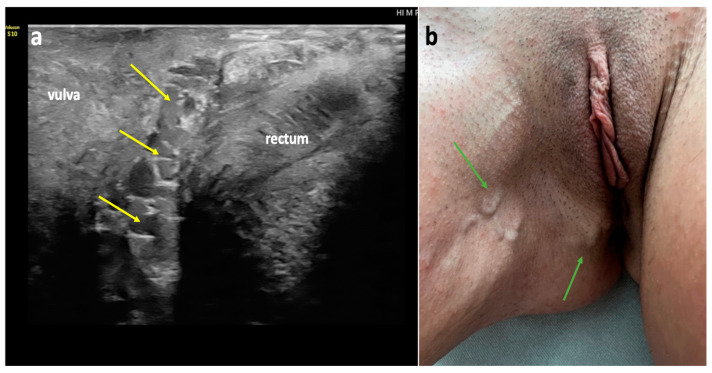
Perineal escape point in a female patient; (**a**) duplex ultrasonography (longitudinal view) of an incompetent and dilated perineal vein (yellow arrows), running alongside the posterior wall of the vagina, between the vagina and rectum; (**b**) atypical varicose veins (green arrows) related to incompetent perineal veins.

**Figure 10 diagnostics-15-00245-f010:**
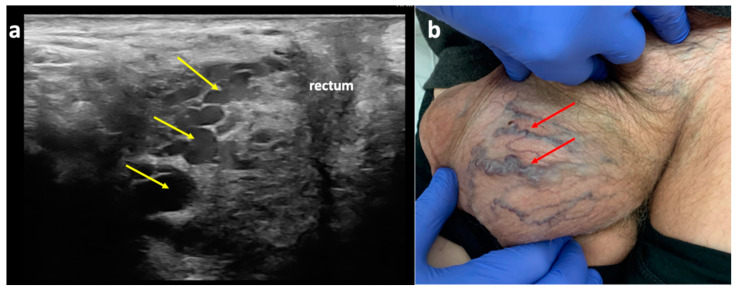
Perineal escape point in a male patient; (**a**) duplex ultrasonography (longitudinal view) of the perineal veins (yellow arrows), revealing their tortuous course anteriorly from the rectum, where they join the internal pudendal veins; (**b**) varicose veins of the scrotum (red arrows) related to incompetent posterior scrotal veins, tributaries of the perineal veins.

**Figure 11 diagnostics-15-00245-f011:**
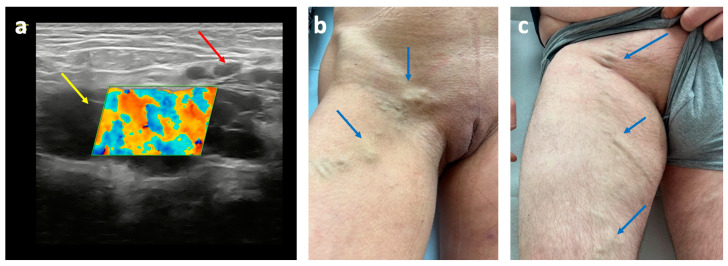
The inguinal pelvic escape point: (**a**) duplex ultrasonography of incompetent round ligament vein (yellow arrow) in the inguinal canal, connected to the external pudendal vein (red arrow); (**b**) extra-pelvic varicose veins of pelvic origin (blue arrows) related to the inguinal pelvic escape point in female patient; (**c**) male patient.

**Figure 12 diagnostics-15-00245-f012:**
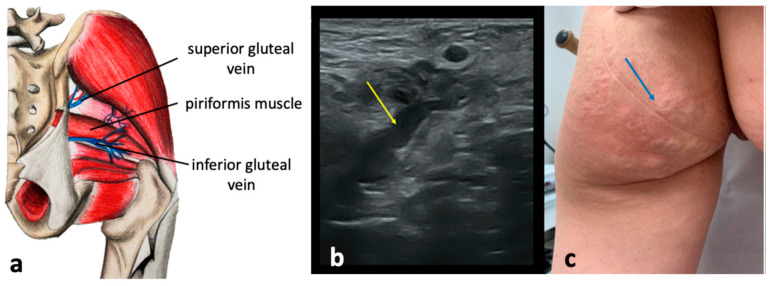
Anatomical scheme (**a**) of the gluteal pelvic escape points related to the superior and inferior gluteal veins; (**b**) ultrasonographic view of the superior gluteal pelvic escape point (yellow arrow) located above the piriformis muscle; (**c**) extra-pelvic varicose veins of pelvic origin (blue arrow) associated with the superior and inferior gluteal escape points.

**Figure 13 diagnostics-15-00245-f013:**
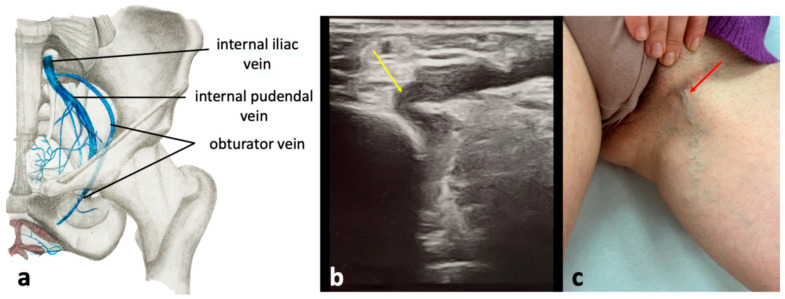
The obturator pelvic escape point; (**a**) anatomical scheme of the topography of the obturator vein; (**b**) ultrasonographic view of obturator pelvic escape point (yellow arrow); (**c**) extra-pelvic varicose veins of pelvic origin (red arrow) related to the obturator pelvic escape point.

## Data Availability

No new data were created or analyzed in this study. Data sharing is not applicable to this article.

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
