# Peer review of "Anatomical, Pathophysiological, and Clinical Aspects of Extra-Pelvic Varicose Veins of Pelvic Origin"

_diagnostics, 2025, doi:10.3390/diagnostics15030245_

Round 1

Reviewer 1 Report

Comments and Suggestions for Authors

Congratulations,

This is an overall interesting and clinicaly important review, which however should better discuss some particular issues (for details see below). Authors may include few additional references relevant for this topic. 

Actually, there is no "extra-pelvic presentation of the pelvic venous disorder" (the title and others, e.g. line 30-31), as there is no separate venous system for different locations, although (as mentioned in lines 147-150) the pelvic and lower limb compartments can easily be distinguished, . Authors should slightly expand and better discuss this issue.  

The term "atypical VV" (line 88) is a bit misleading as the anatomy and pathophysiology of these varicose veins is rather typical, when considering their "pelvic origin".

The real occurence of pelvic symptoms (line 80) is much higher as they presumably are underreported either by patients and doctors due to some embarrassment (compare Health Technol Assess 2016,20:1–108; DOI:10.3310/hta20050 and JCM 2024,13:5053; DOI:10.3390/jcm13175053).

The role of abnormal anatomy (lines 54 and 86-88) is the second, after the number of pregancies (or rather deliveries), causative factor in the women's pelvic veins insufficiency (compare JCM 2021,10:736; DOI:10.3390/jcm10040736; Diagnostics 2022,12:2337; DOI:10.3390/diagnostics12102337). 

In men the role of extremal excercise (i.e. excessive load) seems to be crucial. Therefore, it is disputable, whether PVI in women and men is really the same disease (line 41). Authors should at least shortly comment this issue.

The clinically overt (symptomatic) nut-cracker syndorme is relatively rare. In fact, the renal vein compression (and the impairment of blood outflow with the involvement of some extraperitoneal collateralls) may be due to small angle between aorta and superior mesenteric artery (superior mesenteric aorta in line 132 is the editorial error, I guess), but also may be enhanced only temporarily (e.g. during pregnancy or after significant weight change). Authors should also mention some other anatomic LRV abnormalities, including LRV hypoplasia, more common retroaortic location of the LRV or the left ovarian vein with a drainage to the retraortic branch of LRV 

The preliminary diagnostics of PVI involves mainly ultrasonography (line 90), which however should be further verified by venography in computed tomography or magnetic resonance imaging. Please, mention it shortly.

Author Response

Response to the remarks of the Reviewer 1:

  1. “Actually, there is no "extra-pelvic presentation of the pelvic venous disorder" (the title and others, e.g. line 30-31), as there is no separate venous system for different locations, although (as mentioned in lines 147-150) the pelvic and lower limb compartments can easily be distinguished. Authors should slightly expand and better discuss this issue.”  

            Thank you very much for the comment, although we do not agree with the reviewer. The term “extra- pelvic varicose veins of pelvic origin” is one of the clinical presentations of pelvic venous disorders (PeVD) and has been taken from the SVP (symptoms, varices, pathophysiology) classification of PeVD*. It is classified as:  S3 (extrapelvic symptoms of venous origin) V3 (pelvic origin extrapelvic varices) PPELV, R, NT (pelvic escape points, reflux, nonthrombotic). There is also a chapter dedicated to pelvic venous disorders causing varicose veins in ESVS 2022 Clinical Practice guidelines**. To satisfy the reviewer we decided to add:

“Based on the SVP (Symptoms, Varices, Pathophysiology) classification of PeVD it represents S3 V3 PPELV,R,NT.”

* Meissner MH, Khilnani NM, Labropoulos N, Gasparis AP, Gibson K, Greiner M et al. The Symptoms-Varices-Pathophysiology classification of pelvic venous disorders: A report of the American Vein & Lymphatic Society International Working Group on Pelvic Venous Disorders. J Vasc Surg Venous Lymphat Disord. 2021;9:568-584.

** De Maeseneer MG, Kakkos SK, Aherne T, Baekgaard N, Black S, Bloomgren L et al. Editor's Choice - European Society for Vascular Surgery (ESVS) 2022 Clinical Practice Guidelines on the Management of Chronic Venous Disease of the Lower Limbs. Eur J Vasc Endovasc Surg. 2022;63:184-267.

  1. “The term "atypical VV" (line 88) is a bit misleading as the anatomy and pathophysiology of these varicose veins is rather typical, when considering their "pelvic origin".

It has been changed to “extra-pelvic VVs of pelvic origin”.

  1. “The real occurence of pelvic symptoms (line 80) is much higher as they presumably are underreported either by patients and doctors due to some embarrassment (compare Health Technol Assess 2016,20:1–108; DOI:10.3310/hta20050 and JCM 2024,13:5053; DOI:10.3390/jcm13175053).”

We agree with the reviewer that the prevalence of pelvic symptoms, especially chronic pelvic pain caused by pelvic vein incompetence is much higher and may be as high as 30%. In our paper we mentioned the occurrence of pelvic symptoms in the subgroup of patients with extra-pelvic VVs of pelvic origin, which based on the literature is less than 10%. Based on Gibson study only 7% of patients (5 patients from 72 patients with VVs of pelvic origin) complained of symptoms consistent with pelvic congestion syndrome (such as pain or heaviness in the pelvis). The lower prevalence of pelvic symptoms in patients with VVs of pelvic origin is explained by Meissner*. This occurs due to the transmission of pelvic venous hypertension to distal areas, such as the veins of the vulva and lower limb.

In the reference cited by the reviewer (Health Technol Assess 2016,20:1–108; DOI:10.3310/hta20050), the relationship between pelvic vein incompetence and chronic pelvic pain (CPP) has been mentioned, but nothing has been mentioned about the prevalence of CPP in patients with pelvic origin VVs at the lower limb. Another study cited by the reviewer (JCM 2024,13:5053; DOI:10.3390/jcm13175053) only mentioned that the abdominal and/or pelvic pain or discomfort was significantly higher in the Intact group (non-treated patients), whereas the mean leg symptoms—pain/heaviness and visual scores were significantly higher (worse) in patients from the ReVD group (patients with recurrent venous disease), but the exact numbers and percentages were not presented.

To make it more clear to the reader we decided to slightly modify the sentence to:

“Based on the literature, among patients presenting with extra- pelvic VVs pelvic origin, less than 10% have been reported to have pelvic symptoms related with PVI, as the pelvic venous hypertension is transmitted from pelvis to distal areas, such as the veins of the vulva and lower limb.”

*Meissner MH, Gloviczki P. Chapter 21 - Pelvic Venous Disorders. Atlas of Endovascular Venous Surgery. 2019, Pages 567-599

  1. “The role of abnormal anatomy (lines 54 and 86-88) is the second, after the number of pregancies (or rather deliveries), causative factor in the women's pelvic veins insufficiency (compare JCM 2021,10:736; DOI:10.3390/jcm10040736; Diagnostics 2022,12:2337; DOI:10.3390/diagnostics12102337). “

We edited the paragraph based on the reviewer suggestions:

“This rate is notably lower than that observed in females, potentially indicating pregnancy-related, hormonal, anatomical, or genetic predispositions in the latter.”

and

“Pelvic vein incompetence (PVI) is influenced by several predisposing factors. One of the most important factors is pregnancy. PVI is still rare during the first pregnancy. It typically emerges around the fifth month of the second pregnancy, with an increased risk during subsequent pregnancies. Elevated levels of estrogen, progesterone and relaxin during pregnancy are thought to weaken vein walls and cause dilation [8-10]. Additionally, an increased blood volume and up to a 60-fold rise of the ovarian vein vascular capacity, which may persist for months after delivery may play a role. Compression of the pelvic veins by pregnant uterus can further contribute to this pathomechanism [11]. Additionally, anatomical variations of venous valve distribution may play a role in the development of PVI. The number of valves is higher in distally located veins. Of note, valves are found only in 10% of the IIVs. In the ovarian veins, valves are primarily located in their distal third, while 15% of the left and 6% of the right ovarian veins are devoid of the valves [7].”

  1. In men the role of extremal excercise (i.e. excessive load) seems to be crucial. Therefore, it is disputable, whether PVI in women and men is really the same disease (line 41). Authors should at least shortly comment this issue.”

Thank you for your comment. It has been added:

“Although the prevalence of scrotal varicose veins (VVs) is not well-documented in the literature, some studies have reported a 3% incidence of lower limb VVs of pelvic origin among male patients [4]. This rate is notably lower than that observed in females, potentially indicating pregnancy-related, hormonal, anatomical, or genetic predispositions in the latter. In males, however, the role of extreme exercise (e.g., excessive physical load) appears to be a critical contributing factor. Therefore, it is disputable, whether PVI in women and men is really the same disease.”

  1. “The clinically overt (symptomatic) nut-cracker syndrome is relatively rare. In fact, the renal vein compression (and the impairment of blood outflow with the involvement of some extraperitoneal collateralls) may be due to small angle between aorta and superior mesenteric artery (superior mesenteric aorta in line 132 is the editorial error, I guess), but also may be enhanced only temporarily (e.g. during pregnancy or after significant weight change). Authors should also mention some other anatomic LRV abnormalities, including LRV hypoplasia, more common retroaortic location of the LRV or the left ovarian vein with a drainage to the retraortic branch of LRV “

Thank you for noticing the word error. It has been amended.

Because the real left renal vein compression (Nutcracker) syndrome is relatively rare, as the reviewer mentioned, and in our paper we wanted to be more focused on the connections between pelvic and extra- pelvic veins (pelvic escape points), at the beginning we decided not to go into details of the anatomical variants of the left renal vein. Still, we followed the reviewer suggestions and added:

“This compression can result from a small angle between the aorta and the superior mesenteric artery, but can also be temporary, such as during pregnancy or following a significant weight loss. Other anatomical abnormalities of the LRV, including a rare LRV hypoplasia, or more common retroaortic course of the LRV or the left ovarian vein draining into the retroaortic branch of the LRV, should also be considered.”

  1. The preliminary diagnostics of PVI involves mainly ultrasonography (line 90), which however should be further verified by venography in computed tomography or magnetic resonance imaging. Please, mention it shortly.”

In our paper we focused on the diagnostic of extra- pelvic VVs of pelvic origin. In this case, performing cross sectional imaging is not necessary. According to the European Society for Vascular Surgery (ESVS) guidelines*, as well as U.S. guidelines**, whenever VVs of pelvic origin are suspected, full duplex ultrasound (DUS) of both the lower extremity veins and pelvic escape points is recommended.  According to the ESVS guidelines, additional abdominal and/or transvaginal DUS evaluation should be considered, whereas US guidelines do not advocate this if the patient does not present with pelvic symptoms. Both guidelines support the strategy of a direct, so called ‘bottom-up’ treatment of the pelvic escape points and VVs of pelvic origin, without pelvic vein treatment in patients without pelvic symptoms.

Cross sectional imaging by magnetic resonance venography or computed tomography is recommended in addition to duplex ultrasound assessment when pelvic vein intervention is considered, thus in the patients with pelvic symptoms, especially those with suspected supra-inguinal venous obstruction.

We added the following paragraph:

“In patients with coexisting pelvic symptoms, possibly of venous etiology, when a pelvic vein intervention is considered, the initial diagnosis of PVI, including ultrasonography, should be further verified by detailed imaging, using either magnetic resonance venography or computed tomography [6].”

* De Maeseneer, M.G.; Kakkos, S.K.; Aherne, T.; Baekgaard, N.; Black, S.; Blomgren, L.; et al. Editor's Choice - European Society for Vascular Surgery (ESVS) 2022 clinical practice guidelines on the management of chronic venous disease of the lower limbs. Eur J Vasc Endovasc Surg 2022, 63, 184-267.

** Gloviczki P, Lawrence PF, Wasan SM, Meissner MH, Almeida J, et al. The 2023 Society for Vascular Surgery, American Venous Forum, and American Vein and Lymphatic Society clinical practice guidelines for the management of varicose veins of the lower extremities. Part II: Endorsed by the Society of Interventional Radiology and the Society for Vascular Medicine. J Vasc Surg Venous Lymphat Disord. 2023:101670

Reviewer 2 Report

Comments and Suggestions for Authors

Dear Authors, Thanks for this great work!

It is very important to make the effort again and again to emphasize on these escape points, the participation of pelvic refluxes in leg varicose veins (even without pelvic congestion, as highlighted) and the importance of diagnosing them in order to have a better treatment approach.

As it was Franceschi who first described them together with Bahnini and catalogued the names PP, IP, CP, etc, (all those you mentioned), I would feel it a fair move to give this little approach to the short history about the knowledge of the points and honor the man that described them. 

Good English, very good and practical explanations of the implication of each point. 

Only one comment: 

Lines 40-41

which is nonetheless a lower rate in comparison with females, suggesting an 40 anatomical, hormonal, or genetic predisposition. 

Please consider mentioning pregnancy as a risk factor. 

Author Response

Times New Roman
